# The Role of Greener Innovations in Promoting Financial Inclusion to Achieve Carbon Neutrality: An Integrative Review

Mohsen Brahmi [1,*], Luca Esposito [2,3], Anna Parziale [4], Karambir Singh Dhayal [5], Shruti Agrawal [6], Arun Kumar Giri [5] and Nguyen Thi Loan [7]

1 Department of Economics, University of Sfax, Sfax 2100, Tunisia
2 Karelian Institute, University of Eastern Finland, 80130 Joensuu, Finland; luca.esposito@uef.fi or lucaesposito1@unisa.it
3 Department of Economics and Statistics, University of Salerno, 84084 Fisciano, Italy
4 Department of Business Sciences-Management & Innovation Systems, University of Salerno, 84084 Fisciano, Italy; aparziale@unisa.it
5 Department of Economics and Finance, Birla Institute of Technology & Science (BITS), Pilani Campus, Rajasthan 333031, India; p20210068@pilani.bits-pilani.ac.in (K.S.D.); akgiri.bits@gmail.com (A.K.G.)
6 Department of Humanities and Social Sciences, Malaviya National Institute of Technology Jaipur, Rajasthan 302017, India; 2019rhs9506@mnit.ac.in
7 Department of Business Administration, Faculty of Economic and Business Administration, Hong Duc University, Thanh Hóa 440000, Vietnam; nguyenloan@hdu.edu.vn
* Correspondence: brahmi.mohsen@gmail.com

**Abstract:** In recent times, the green transition, by promoting carbon neutrality, has become highly imperative to meet environmental challenges. The present literature review study seeks to explore the intersecting role of greener innovations in facilitating financial inclusion for a sustainable future. Within the global agenda is the goal of carbon neutrality, with the aim of reducing environmental impact and mitigating climate change. *Aim:* The present study aims to investigate the role that technological innovations play in the financial inclusion of achieving climate neutrality. *Method:* Through a systematic literature review, we investigate how new innovations generate new investment opportunities and promote sustainable development. However, fair, accessible, and inclusive financing is crucial. *Findings:* the analyzed documents in this study shows that technological innovations can play an important role in financial inclusion for carbon neutrality and provide some important policy implications. Indeed, a favorable regulatory environment could generate positive effects already in the short term, with important environmental, economic, and social repercussions.

**Keywords:** carbon neutrality; greener innovations; mitigating climate change; financial inclusion; sustainable development; economic





## 1. Introduction

The problem of the climate emergency is central to both national and international political agendas. As the green transition gains momentum, there is an unprecedented chance to make the structural changes required for a significant drop in global emissions, which must be taken advantage of (Stern and Valero 2021). Companies are urged to decrease their own direct and indirect emissions as matters of a long-term strategic business decision and the ethical responsibility to safeguard the environment and future generations.

The issue of access to finance and new capital is becoming increasingly relevant with respect to current developments and sustainability. In fact, virtuous companies in these areas can become more attractive and resilient for the markets and investors. Proof of this is the fact that Environmental, Social, and Governance (ESG) funds are growing both at an equity and debt level (according to the most recent analyses, they protect investors from systemic risk). Inclusive finance, therefore, has a key role to play in supporting companies not only in mitigation actions with respect to climate change

but also in adaptation, taking into consideration the fundamental objective of protecting biodiversity and natural resources, which can be allies in the reduction of emissions (a priority area also in the European Taxonomy), as well as the promotion of the transition to a circular economy (Zhai et al. 2022).

The term "financial inclusion" refers to the provision of affordable, adequate, and effective financial services to different groups (Anu et al. 2023). Although some countries may have seen a rise in access to financial inclusion, the rate and depth of that development vary greatly among them. When financial inclusion is promoted, small companies can now access credit that was previously inaccessible through conventional financial channels, boosting productivity and quickening development. (Yao et al. 2023). Currently, according to various estimates and research, the world is proceeding rather slowly in the process of decarbonization, above all, due to the lack of efficient technologies. Therefore, the need to push the accelerator of technological innovation in general and green innovation is now clear (Nassani et al. 2023).

Green technology innovation is a generic term for new technologies, new processes, and new products that help reduce environmental pollution and save resources. Therefore, it is one of the key pathways for heavily polluting businesses to achieve the green transition. In this sense, green innovation has thus been acknowledged as one of the major forces affecting economic development, environmental sustainability, and quality of life (Bai and Lyu 2023). It is the development of new technology or software that supports the use of environmentally friendly goods or procedures. This category of innovation goes beyond regulatory compliance and includes technological advancements in energy conservation, pollution prevention, waste recycling, green product design, and corporate environmental management (Wang et al. 2022).

Thus, companies involved in a process of continuous change and development that frequently yields observable green developments are considered to be green innovative firms. These factors help to explain why green innovation has lately become a significant driver of sustainable development. Companies view it as a means to both meet consumer and regulatory demands and improve efficiency by making better use of natural resources. By recycling and reducing waste, green innovation lowers costs while differentiating goods on the market to appeal to environmentally conscious customers (Yousaf 2021). However, green technology innovation requires stable and continuous funds. There is a high degree of uncertainty, leading to the problem of severe financial constraints for heavily polluting enterprises to implement green technology innovation (Takalo et al. 2021; Xue and Zhang 2022). Research and development efforts for dedicated technologies must be supported by institutions, in collaboration with the production and financial system, promoting related investment capacity, "open" innovation, and a partnership-oriented vision (Li and Lu 2023).

Dhayal et al. (2023) mentions the role of green venture capital (GVC) in businesses and companies that aim to make a difference for environmental protection, the present research aims to explore green innovations occurring in the space of financial inclusiona as a key factor that can strongly contribute to the decarbonization process.

The methodology employed was bibliometric analysis and a systematic literature review (SLR) research approach of analyzing and synthesizing published articles to identify the emerging and trending research opportunities in green innovation and financial inclusion. This paper provides a valuable contribution to fill an important gap in the literature by outlining a clear framework and a detailed, comprehensive clear overview of the existing literature on our research topic. The implications and conclusions of our research highlight important insights for policymakers and researchers.

The paper is structured as follows. After a literature review in the following section, Section 3 describes the research methodology, and in Section 4, we describe the results obtained and show the findings. Section 5 presents the conclusions, policy implications, and future research directions.

## 2. Literature Review

The financial sector is vital for economies to perform well. The challenge of climate change has impacted it also (IEA 2021a, 2021b). Now, governments and policymakers have been trying to tackle the issue of climate change through the financial sector by deploying green finance for future sustainable development (IPCC 2018). Green fintech has gained a lot of traction due to its emphasis on sustainability by achieving carbon neutrality through financial inclusion with the use of technologies that have the ability to lower carbon emissions to a great extent compared to traditional banking services.

Thus, firms, by simply incorporating green finance into their operational activities, have been ensuring financial service access to everyone for a better equitable society where environmental protection is of prime importance. Thus, green financial inclusion has been boosting economic growth and preventing environmental degradation as well. It offers many incentives for businesses to invest in making a positive impact on the environment and society. It has been realized that a lot of green innovation investments in the financial sector have been happening through green venture capital (Dhayal et al. 2023), green fintech (Muganyi et al. 2021), green financial inclusion (Liu et al. 2022a, 2022b), etc.

The present study aims to explore these innovations in the financial inclusion space, which can contribute towards decarbonization. Climate change has been leading to severe weather conditions, which are quite scary. It poses a risk offood scarcity and intensifies poverty, health risks, and societal conflicts (IPCC 2023). These impacts are causing more damage to vulnerable low-income populations. Countries have been trying to mitigate and build resilience tothe effects seen on the environmental, health, and social dimensions of climate change (UNEP 2022). Studies have shown that GFI builds resilience at the individual and collective levels to the effects of climate change (Ahmad et al. 2022; Deng et al. 2019; Dhayal et al. 2023; Dong et al. 2022; Hinson et al. 2019; Le et al. 2019; Liu et al. 2022a, 2022b; Muganyi et al. 2021; Qin et al. 2021).

Green finance and financial inclusion were considered to be two mutually exclusive concepts. Recently, a lot of overlap has been found between these two different areas. It has been realized that the target population of financial inclusion schemes is also one of the most vulnerable groups, which has to face the brunt of climate change impact. This holistic approach is considered to be green financial inclusion. Green financial inclusion (GFI) has been gaining attention, especially among vulnerable communities and countries. It has the potential to boost climate resilience by boosting the microfinancing of small-scale borrowers who do not have to face the impact of climate change.

The main objective of GFI is to ensure the development of a cleaner environment through carbon neutrality. GFI reduces carbon footprints in several ways. The need for physical travel-based transactions has been reduced due to mobile banking technologies, making it quite affordable and accessible for even people situated in remote locations (Yu et al. 2022). The fast disbursal of loans to SMEs helps sustainable projects and farmers in making shifts to climate-smart agriculture. GFI has also given rise to impact investing, where organizations are generating returns beyond the notion of profit-making.

Therefore, GFI reduces carbon emissions by encouraging the flow of investments in renewable energy, promoting the use of energy-efficient technologies, and supporting sustainable projects in different sectors of the economy. Thus, GFI can help create a greener future, which will be carbon-neutral, and there will be access to projects that are predicted to further reduce the carbon footprint through the use of renewable energy in financial transactions.

## 3. Research Methodology

In the present article, we have applied the bibliometric analysis and systematic literature review (SLR) research approach to provide a detailed, extensive overview of existing studies. To conduct the thematic overview of the existing literature on green innovation and financial inclusion, we used bibliometric analysis. It is one of the most comprehensive approaches for tracing the wealth of information on the study topic (Ellegaard and Wallin 2015). SLR is an appropriate methodology to recognize the emerging and trending

research scopes in a particular area of research, and thus, to analyze and summarize the existing articles.

Although SLR is defined as "an efficient technique for hypothesis testing, for summarizing the results of existing studies, and for assessing consistency among previous studies; these tasks are clearly unique to medicine" (Paul et al. 2021), in the recent past, several bibliometric and SLR methodologies have been adopted in various research areas by using different search databases (Han et al. 2019; Paul and Barari 2022; Donthu et al. 2021).

### 3.1. Research Questions

We formulated three research questions in the present study, which are as follows:

RQ1: What is the research publication trend in this area?
RQ2:What are the influential articles, authors, most productive journals, most contributing countries, and affiliations in this area?
RQ3: What is the conceptual structure in this area?

### 3.2. Search Strategy

To extract the relevant articles, the search strategy was implemented in March 2023 for articles related to green innovation and financial inclusion. The articles were extracted from the Scopus database since the Scopus and Web of Science (WoS) databases provide an advantage for applying bibliometric methods, which prompted the authors (Paul et al. 2021).

Scopus covers the majority of scholarly articles, which helps to create future opportunities for researchers who intend to conduct research on a particular research theme. Scopus has more extensive coverage than the other available databases (Farooque et al. 2019). It is the largest database of abstracts and citations of peer-reviewed research articles globally (Farooque et al. 2019). Table 1 presents the search strings using keywords related to the research.

We applied the following inclusion criteria.

1.  All of the documents are in the English language;
2.  The time duration for the articles was from 2013 to March 2023;
3.  All of the documents are from peer-reviewed journals;
4.  All of the documents are focused on the area of green innovation and financial inclusion;
5.  The documents are in the short or full version (not an editorial or abstract).

**Table 1.** Search string used for the present study.

| Database | Scopus |
|---|---|
| Keywords | "Financial Inclusion" OR "Fintech" OR "Green Fintech" OR "Digital Financial Inclusion" OR "Green Financial Inclusion" OR "Digital Finance" OR "Digital Payment" <br> **AND** <br> "Green Finance" OR "Green Innovation" OR "Sustainable" OR "Carbon Neutrality" OR "Renewable Energy" OR "Climate Change" OR "Energy Efficiency" OR "Sustainable Development" OR "Carbon Dioxide" |
| Timespan | 2013 to 2023 (until March) |

Furthermore, to remove the unbiasedness from the study, the abstracts were read by three authors, and irrelevant articles were removed. Finally, 290 articles were considered for the final review. The procedure for the extraction of the articles is presented in Figure 1.

### 3.3. Data Analysis

To conduct the analysis of the final 290 extracted articles, the authors used the Bibliometrix package and VOS-viewer for bibliometric analysis and SLR. The analysis of the existing literature on green innovation and financial inclusion focused on: (1) yearly

publications; (2) influential authors; articles and institutes; sources; (3) frequently used keywords; (4) citation and co-citation analysis.

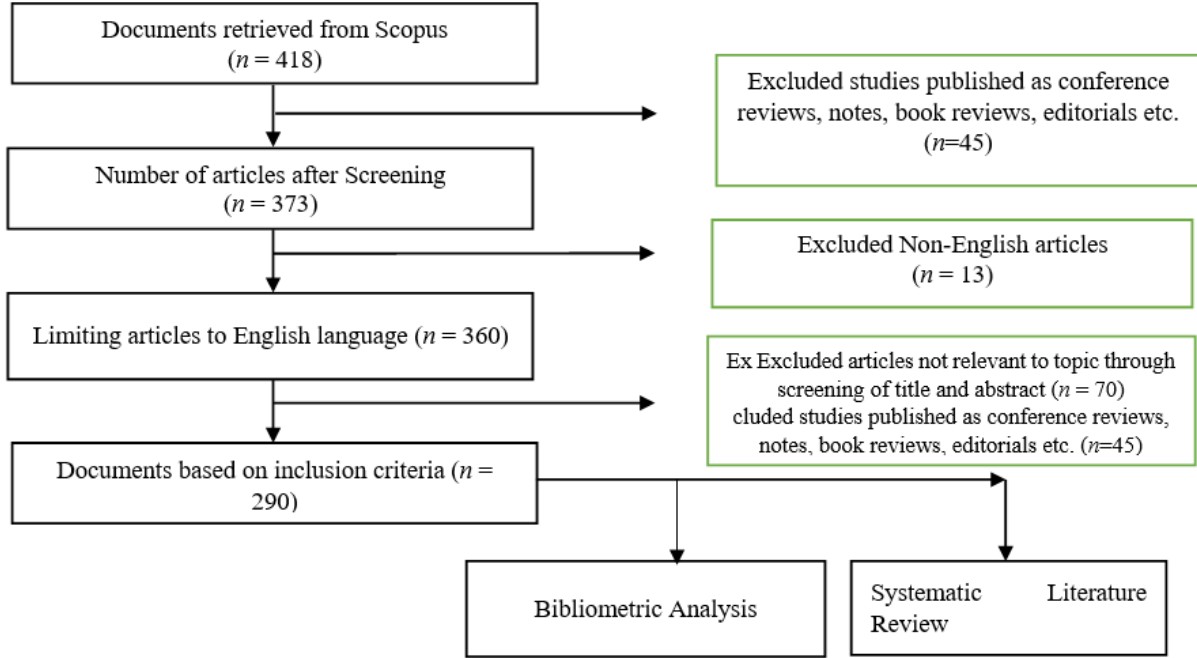

**Figure 1.** Extraction of articles and selection process.

## 4. Results

### *4.1. Main Information*

In earlier research, authors have employed numerous techniques for bibliometric analysis, each with its own set of advantages and disadvantages. Here, we categorized the information of the bibliometric analysis into three parts, consisting of the bibliographic data, the author's information, and the document content. This provided information about the total number of authors in this research area, the authors working in collaboration, and the types of articles published (refer Table 2).

**Table 2.** Summary of the bibliometric analysis.

| Main Information about Data | |
|---|---|
| **Timespan** | **2013 to 2023** |
| Sources | 113 |
| Total number of publications | 290 |
| Annual growth rate % | 38.52 |
| Document average age | 1.85 |
| Average citations per doc | 10.05 |
| References | 18,655 |
| **Document Information** | |
| Keywords plus (ID) | 742 |
| Author's keywords (DE) | 940 |
| Article | 272 |
| Review | 18 |
| Author information | |

**Table 2.** *Cont.*

| Main Information about Data | |
| --- | --- |
| **Authors** | 833 |
| Authors of single-authored docs | 30 |
| Single-authored docs | 32 |
| Co-authors per doc | 3.27 |
| International co-authorships % | 29.31 |

Source: Authors' contribution.

### 4.2. Year-Wise Publication Progress

Green innovation in financial inclusion to achieve carbon neutrality is an emerging research area in economics. In the past few years, articles related to green innovation and carbon neutrality issues in sustainability have indicated several future research scopes. It can be seen in Figure 2 that the number of publications in this domain has been increasing rapidly since 2019. It can be noticed that fewer articles were published in this area in during 2013 to 2018. The number of publications has grown over the previous four years, indicating that scholars' interest is expanding in green innovation, financial inclusion, and sustainability. The reason behind the sudden increase could be the United Nations Sustainable Development Goals 2015, which focus on sustainability and environmentally positive actions.

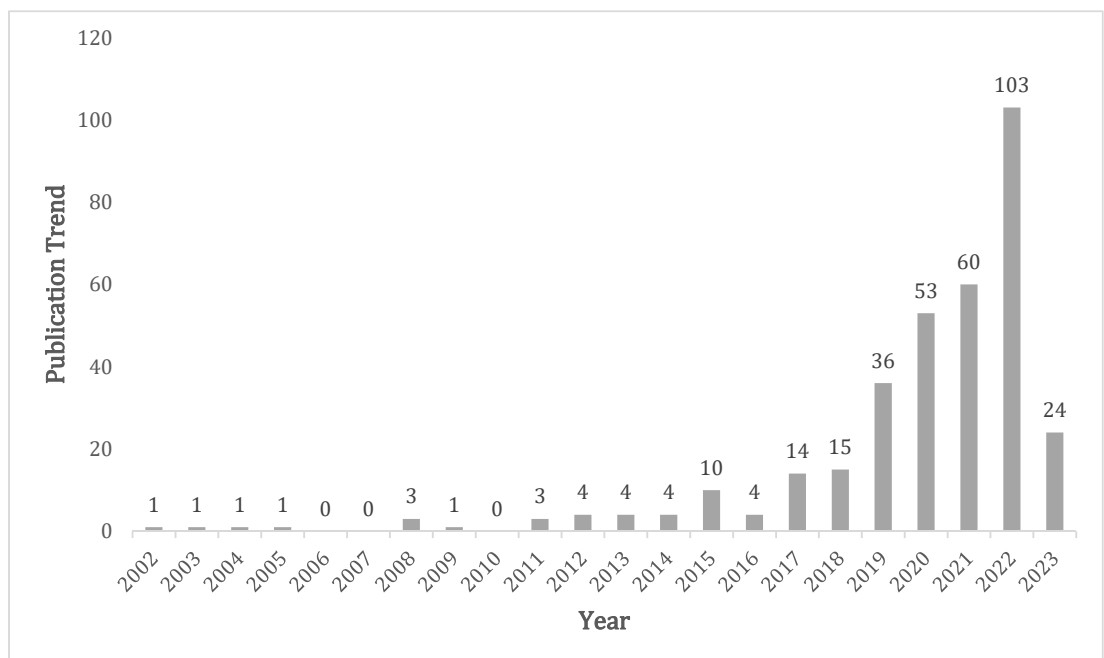

**Figure 2.** Yearly publications.

### 4.3. Most Influential Articles

To answer RQ2, the data of the most influential articles were collected from the Scopus database and imported into R package for analysis. The top 25 contributing articles in the field of green innovation and financial inclusion are presented in Table 3. It can be seen in Table 3 that the articles published by Usman et al. (2021), Le et al. (2020), and Muganyi et al. (2021) were the top contributing articles, with total citations (TC) of 246, 200, and 94, respectively, in the fields of green innovation and financial inclusion. Usman et al. (2021) explored the determinants of ecological footprints and economic growth to identify the progress of financial development and the utilization of renewable and non-renewable energy sources that contribute to reducing carbon footprints for 15 high-emitting countries.

**Table 3.** Summary of top contributing articles.

| No. | Authors | Title | Year | Journal | TC |
|---|---|---|---|---|---|
| 1 | (Usman et al. 2021) | *Do financial inclusion, renewable and non-renewable energy utilization accelerate ecological footprints and economic growth? Fresh evidence from 15 highest emitting countries* | 2021 | *Sustainable Cities and Society* | 246 |
| 2 | (Le et al. 2020) | *Does financial inclusion impact $CO_2$ emissions? Evidence from Asia* | 2020 | *Finance Research Letters* | 200 |
| 3 | (Muganyi et al. 2021) | *Green finance, fintech and environmental protection: Evidence from China* | 2021 | *Environmental Science and Ecotechnology* | 94 |
| 4 | (Le et al. 2019) | *Financial inclusion and its impact on financial efficiency and sustainability: Empirical evidence from Asia* | 2019 | *Borsa Istanbul Review* | 92 |
| 5 | (Zaidi et al. 2021) | *Dynamic linkages between financial inclusion and carbon emissions: Evidence from selected OECD countries* | 2021 | *Resources, Environment and Sustainability* | 90 |
| 6 | (Arner et al. 2019) | *Sustainability, FinTech and Financial Inclusion* | 2019 | *European Business Organization Law Review* | 86 |
| 7 | (Ozturk and Ullah 2022) | *Does digital financial inclusion matter for economic growth and environmental sustainability in OBRI economies? An empirical analysis* | 2022 | *Resources, Conservation and Recycling* | 70 |
| 8 | (Zhou et al. 2022) | *The impact of fintech innovation on green growth in China: Mediating effect of green finance* | 2022 | *Ecological Economics* | 72 |
| 9 | (Pizzi et al. 2021) | *Fintech and SME sustainable business models: Reflections and considerations for a circular economy* | 2021 | *Journal of Cleaner Production* | 66 |
| 10 | (Shahbaz et al. 2022) | *How financial inclusion affects the collaborative reduction of pollutant and carbon emissions: The case of China* | 2022 | *Energy Economics* | 60 |
| 11 | (Qin et al. 2021) | *Does financial inclusion limit carbon dioxide emissions? Analyzing the role of globalization and renewable electricity output* | 2021 | *Sustainable Development* | 54 |
| 12 | (Mehmood 2022) | *Examining the role of financial inclusion towards $CO_2$ emissions: presenting the role of renewable energy and globalization in the context of EKC* | 2022 | *Environmental Science and Pollution Research volume* | 54 |
| 13 | (Hinson et al. 2019) | *Transforming agribusiness in developing countries: SDGs and the role of FinTech* | 2019 | *Current Opinion in Environmental Sustainability* | 46 |
| 14 | (Deng et al. 2019) | *FinTech and sustainable development: Evidence from China based on P2P data* | 2019 | *Sustainability* | 44 |
| 15 | (Ahmad et al. 2022) | *Financial Inclusion, Technological Innovations, and Environmental Quality: Analyzing the Role of Green Openness* | 2022 | *Frontiers in Environmental Science* | 37 |
| 16 | (Cristina and Agudo 2021) | *Fintech and sustainability: Do they affect each other?* | 2021 | *Sustainability* | 35 |
| 17 | (Liu et al. 2022a) | *Impact of Green financing, FinTech, and financial inclusion on energy efficiency* | 2022 | *Environmental Science and Pollution Research* | 30 |

**Table 3.** *Cont.*

| No. | Authors | Title | Year | Journal | TC |
|---|---|---|---|---|---|
| 18 | (Geng and He 2021) | *Digital financial inclusion and sustainable employment: Evidence from countries along the belt and road* | 2021 | *Borsa Istanbul Review* | 26 |
| 19 | (Puschmann et al. 2020) | *How green fintech can alleviate the impact of climate change—The case of Switzerland* | 2020 | *Sustainability* | 21 |
| 20 | (Dong et al. 2022) | *Can financial inclusion facilitate carbon neutrality in China? The role of energy efficiency* | 2022 | *Energy* | 15 |
| 21 | (Liu et al. 2022b) | *Financial inclusion and green economic performance for energy efficiency finance* | 2022 | *Economic Change and Restructuring* | 14 |
| 22 | (Yu et al. 2022) | *Digital finance and renewable energy consumption: evidence from China* | 2022 | *Financial Innovation* | 14 |
| 23 | (Al-Okaily et al. 2021) | *Sustainable fintech innovation orientation: A moderated model* | 2021 | *Sustainability* | 12 |
| 24 | (Feng et al. 2022) | *Financial inclusion and its influence on renewable energy consumption-environmental performance: the role of ICTs in China* | 2022 | *Environmental Science and Pollution Research* | 12 |

Source: Authors' contribution.

Le et al. (2020) have identified the impact of financial inclusion on carbon emission in Asia by constructing three proxy instruments of financial inclusion through principal component analysis and found that income, foreign direct investment, and urbanization led to higher carbon emissions in the regions. Muganyi et al. (2021) examined the impact of green finance policies in their country and concluded that growth in fintech led to the depletion of sulfur dioxide emissions, and thus, had a beneficial impact on the environment. In the top 25 articles, the authors were mostly working in the fields of financial inclusion, sustainability, fintech, carbon emissions, and energy consumption.

### 4.4. Most Influential Authors

To answer RQ2, the data of the most influential articles were extracted from the Scopus database and uploaded into an R package for analysis. Table 4 shows the top ten authors in the fields of green innovation and financial inclusion. It can be seen that Banna H., Sun H., and Wang Y. were the top three contributing authors. The authors have worked in collaboration with many authors and published four articles each, with total citations of 27, 27, and 18, respectively, in the field of green innovation and financial inclusion. The authors have highlighted the role of digital financial inclusion during the COVID-19 pandemic and the impact of financial inclusion on carbon efficiency. It was found that the majority of the top authors working in the area are from social science, energy, and environmental science research backgrounds.

**Table 4.** Top 10 cited authors.

| Author | TP | TC |
|---|---|---|
| Banna H. | 4 | **27** |
| Sun H. | 4 | **27** |
| Wang Y. | 4 | **27** |
| Alam Mr. | 3 | 18 |
| Babajide AA. | 3 | 32 |
| Du M. | 3 | 04 |
| Jorge-vazquez J. | 3 | 40 |
| Li Y. | 3 | 05 |
| Liu Y. | 3 | 06 |
| Moro-visconti R. | 3 | 03 |

Note: TP: total publications; TC: total citations.

### 4.5. Most Productive Journals

In this study, the classification of the journals was extracted from R Studio using the Bibliometrix package to answer RQ2. In the classification of the journals, it was found that *Sustainability (Switzerland)* had the most publications related to green innovation and financial inclusion, with a total of 962 citations, followed by *Frontiers in Environmental Science* and *Economic Research*, with total citations of 70 and 80, respectively. The top 10 journals, with their publication counts, are shown in Table 5. In the analysis, h_index shows the number of articles (h) published in particular journals that have been cited h number of times. Here, g_index shows the weightage given to highly cited journals in the different research articles.

### 4.6. Top Contributing Countries and Affiliations

The scope of green innovation in economics has gained the attention of many academicians and researchers, as reflected by the contributions of authors regarding green innovation from various countries. The top ten contributing countries can be seen in Table 6. In the top 10 countries, China, the United Kingdom, and Spain are the most prominent, with total citations of 708, 297, and 147. The analysis shows that majorly developed countries are working in this area compared to developing countries (answer to RQ2). The top ten affiliations are also shown in Table 6, which shows that the researchers from these institutes are working mostly in this research field. The University of Kinshasa has the most citations,

followed by Covenant University and the Shandong University of Technology with 17, 15, and 11 citations respectively (answer to RQ2).

**Table 5.** Top ten journals.

| Source | TP | h_index | g_index | TC | PY_start |
|---|---|---|---|---|---|
| *Sustainability (Switzerland)* | **85** | 20 | 27 | **962** | 2018 |
| *Frontiers in Environmental Science* | **21** | 4 | 7 | **70** | 2022 |
| *Economic Research-ekonomskaistrazivanja* | **11** | 5 | 8 | **80** | 2020 |
| *Environmental Science and Pollution Research* | **10** | **7** | **10** | 130 | **2022** |
| *International Journal of Environmental Research and Public Health* | 9 | 4 | 6 | 37 | 2021 |
| *Journal of Open Innovation: Technology, Market, and Complexity* | 6 | 5 | 6 | 103 | 2021 |
| *Financial Innovation* | 5 | 4 | 5 | 63 | 2018 |
| *Journal of Islamic Monetary Economics and Finance* | 5 | 3 | 5 | 38 | 2019 |
| *International Journal of Economics and Business Administration* | 3 | 3 | 3 | 15 | 2019 |
| *Review of Development Economics* | 3 | 3 | 3 | 10 | 2020 |

Note: TP: total publications; PY_start: publication year.

**Table 6.** Top countries and affiliations.

| Country | TC | Affiliation | TC |
|---|---|---|---|
| **China** | **708** | **University of Kinshasa** | **17** |
| **United Kingdom** | **297** | **Covenant University** | **15** |
| **Spain** | **147** | **Shandong University of Technology** | **11** |
| **Pakistan** | **146** | **Liaoning University** | **10** |
| Hong Kong | 122 | Wuhan University | 8 |
| Hungary | 119 | Beijing Technology and Business University | 7 |
| Malaysia | 82 | Catholic University of Avila | 7 |
| Japan | 81 | China University of Mining and Technology | 7 |
| South Africa | 76 | Hunan University | 7 |
| Korea | 73 | Northwest A&F University | 7 |

Note: TC: total citations.

### 4.7. Conceptual Structure

Using bibliometric analysis, the conceptual structure can help us understand how the knowledge of a relevant domain is organized and connected among the different ideas and topics. Moreover, it can help us understand the different relationships among the various concepts, themes, and topics that emerge from the various patterns of scholarly publications. In the map method of multiple correspondence analyses (MCA), each point represents a specific idea or concept, and their proximity shows the closeness between them. Thus, it can help us understand closely related concepts that are grouped together in clusters.

To answer RQ3, the conceptual structure in the field of green innovation and financial inclusion can be seen in Figure 3, developed using Biblioshiny tool of Bibliometrix, which enables multiple correspondence analyses (MCA). It helps to construct a conceptual structure in a particular research area and presents the k means clustering that associates the articles' groups and their similar areas (Ejaz et al. 2022). MCA is a multivariate interpretive method for analyzing multivariate categorical data graphically and mathematically (Matute and Linsen 2022).

MCA examines the interconnectedness of a group of categorical variables in order to identify new unknown variables or factors (Ejaz et al. 2022). MCA is a novel and statistically proven method to obtain distinction in a particular research domain. Such statistical

methods minimize the number of research dimensions that result in the formation of two-dimensional visualizations, which highlight research area similarities (Ejaz et al. 2022). In the figure, the terms that are closer to the center of the map and those that are more widely dispersed represent subjects that have captured the attention of researchers in recent times, whereas terms that are evenly scattered are related to those less often used in research topics (Ejaz et al. 2022).

This study also assessed the ability to create a contextual structure map of each phrase that often appears in research publications on the issue of the metaverse in green innovation and financial inclusion based on the mapping of the relationship between one word and another, as shown in Figure 3. Each keyword was placed based on the values of Dim 1 and Dim 2 to create a mapping of keywords whose values are very similar. A total of 80 keywords were divided into two different sections: red and blue, and both groups had additional keywords that explained the theme/s of the research on green innovation and financial inclusion. Figure 3 shows that the red-colored keyword clusters are emerging terms, and they are influenced by the blue-colored clusters. For instance, "investment" influences "technology", "adoption", and "financial management". "Environmental protection" influences "climate change", "energy efficiency", and "carbon".

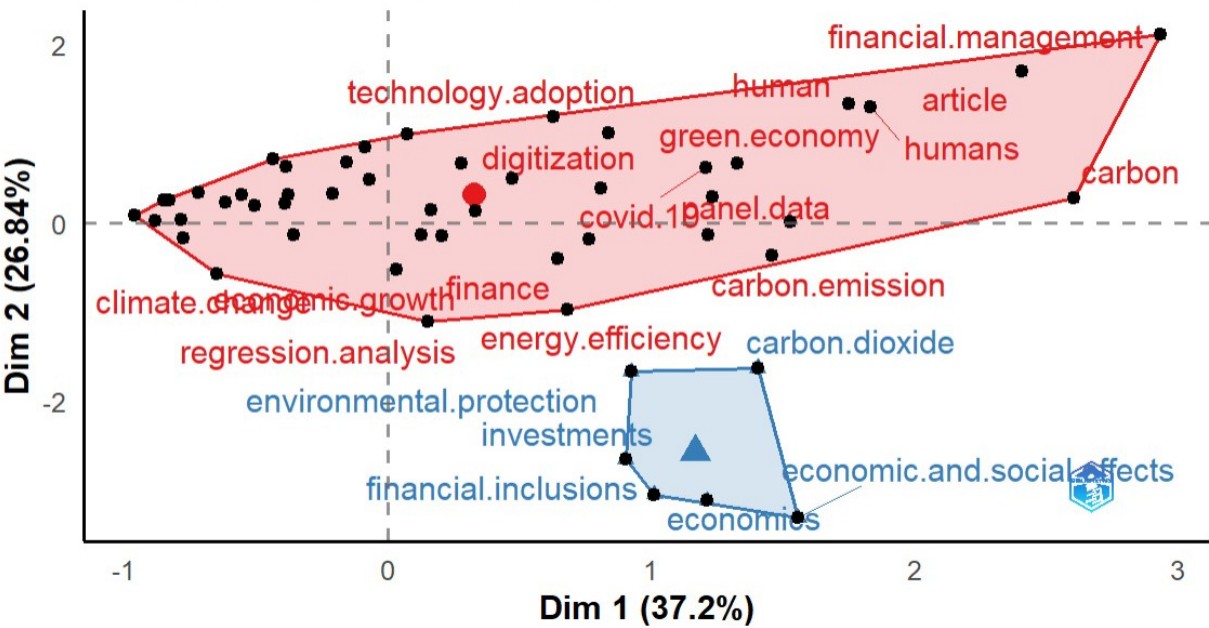

**Figure 3.** Conceptual structure: Map method of multiple correspondence analysis (MCA) of green innovation and financial inclusion. Source: Authors' contribution.

### 4.8. Citation Analysis

In the citation analysis of the articles, the top-cited articles in the area of green finance and financial inclusion were determined using VOS viewer (Figure 4). The citation analysis measured the "degree of connectivity" among various research articles (Tsay 2009). In the top-cited articles, (Arner et al. 2019) had the highest number of citations (i.e., 86), followed by (Zaidi et al. 2021), with 84 citations. These two articles highlight financial technology (fintech) as a major enabler of financial inclusion and include sustainability and the linkage between financial inclusion and energy consumption. The major articles are from the sustainability and financial inclusion research areas, which indicates that financial inclusion has extensive research opportunities in this field of sustainability. It can be seen that the authors of similar colored nodes are working in similar areas of green innovation and financial inclusion and collaborating with different authors of different colored nodes.

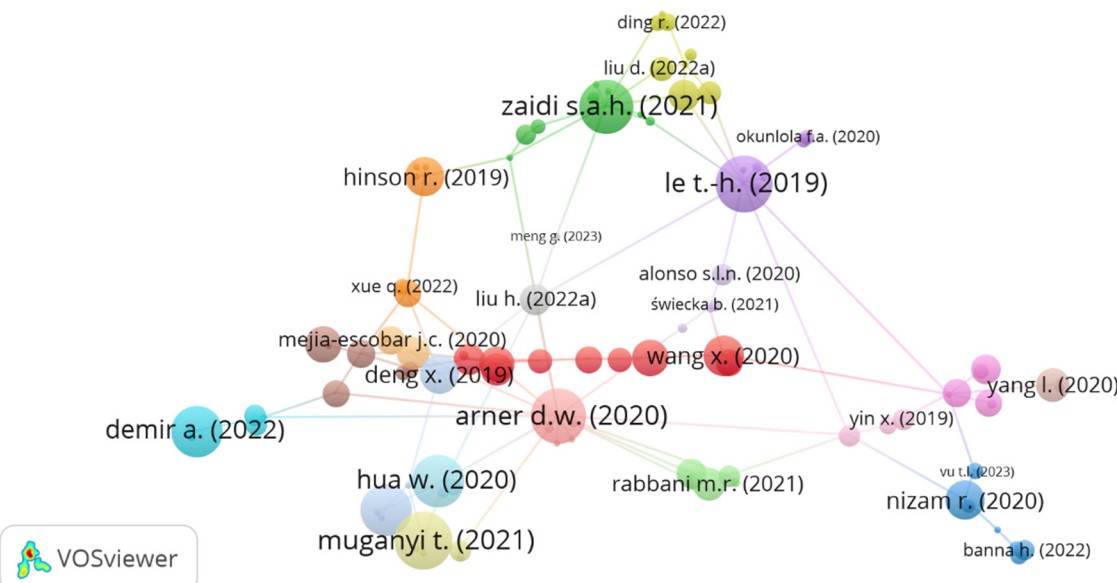

**Figure 4.** Citation analysis of authors. Source: Authors' contribution.

*4.9. Co-Citation Analysis*

Co-citation analysis is the tracking of pairs of research articles that are cited together in source publications. "Co-citation of two research documents occurs when both of the research documents are cited in the third research document" (Boyack and Klavans 2010).

Generally, a co-citation network can be obtained using the formula:

$$Bcocit = article \times citedreferencearticle \tag{1}$$

where *Bcocit*: represents a symmetrical matrix.

The leading papers of each cluster obtained from the cluster analysis are shown in Figure 5. It shows that the authors in the red clusters cite each other, as do those in the green, yellow, and blue clusters. The figure shows that Farhad Taghizadeh-Hesary had the highest number of citations in the blue cluster, with 96 citations, followed by Asli Demirguc-Kunt, with 109 citations in the green cluster, Ross Buckley, with 88 citations in the yellow cluster, and Jinlong Wang, with 89 citations in the red cluster.

*4.10. Most Used Keywords*

The most frequently used keywords in green innovation and financial inclusion articles were extracted using RStudio software (See Figure 6). "Sustainable development" and "China" were the most used keywords (with 65 occurrences). It is followed by "sustainability (50) and "economic development" (48) (See Table 7). In most publications, "sustainable development" was used as the most frequently used keyword, while in some publications, "sustainability" was used as the main keyword. The visualization of the author's keywords from (2013 to 2023) can also be seen in Figure 3. This shows that "innovation", "banking", "finance", and "economic growth" are emerging keywords in this area.

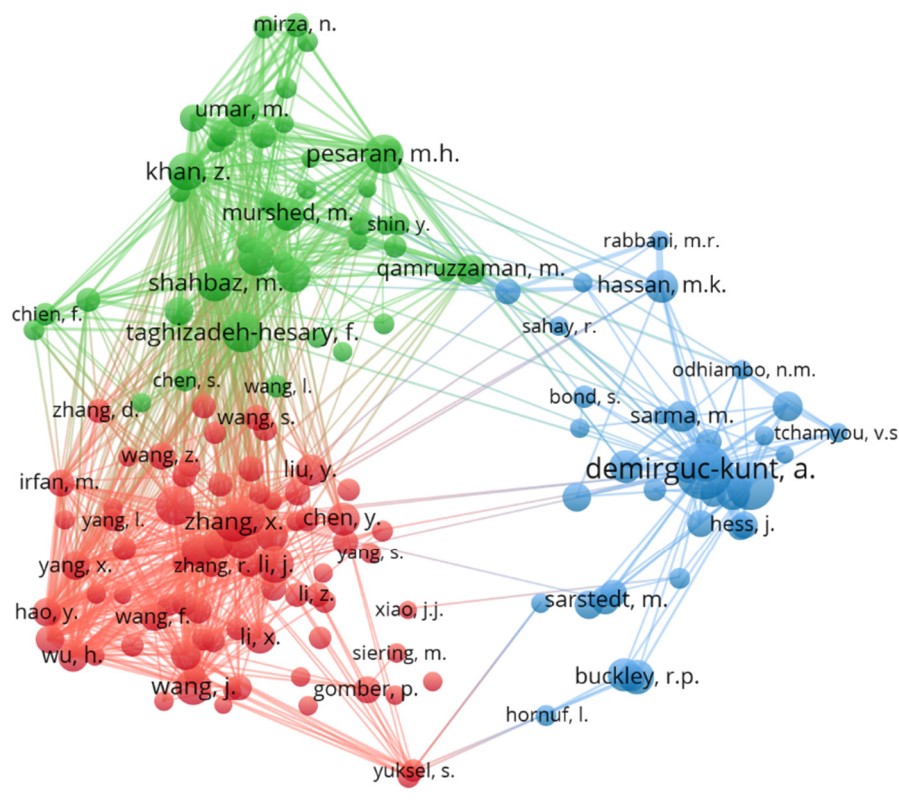

**Figure 5.** Co-citation analysis of authors. Source: Authors' contribution.

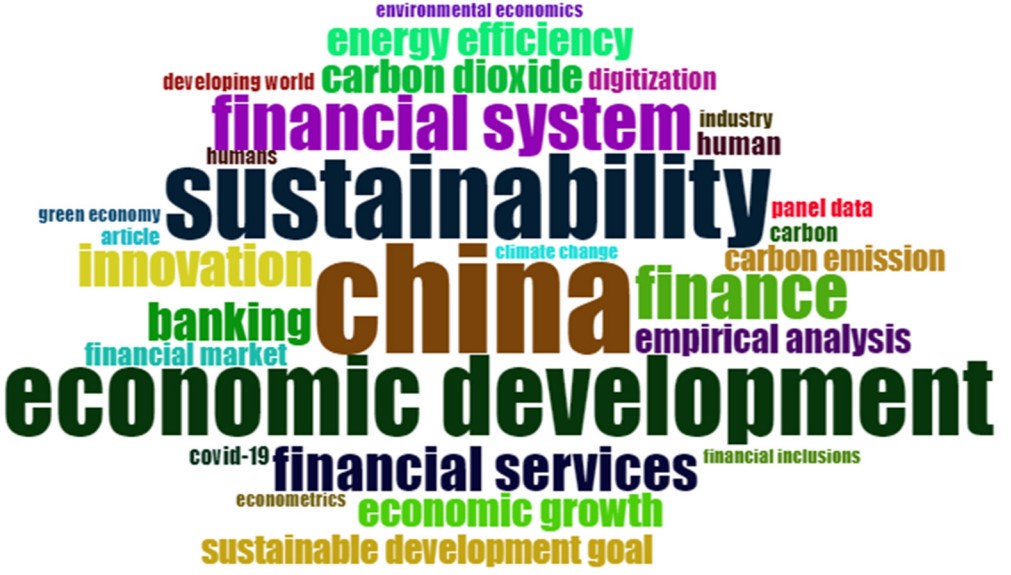

**Figure 6.** Word cloud (2013 to 2023). Source: Authors' contribution.

**Table 7.** Top used keywords.

| Most Frequent Words | | | |
|---|---|---|---|
| **Words** | **Occurrences** | **Words** | **Occurrences** |
| China | 65 | Economic growth | 20 |
| Sustainable development | 65 | Energy efficiency | 20 |
| Sustainability | 50 | Sustainable development goal | 17 |
| Economic development | 48 | Carbon emission | 15 |
| Financial system | 33 | Financial market | 14 |
| Finance | 32 | Digitization | 13 |
| Financial services | 27 | Carbon | 11 |
| Innovation | 25 | COVID-19 | 11 |
| Banking | 23 | Panel data | 11 |
| Carbon dioxide | 20 | Developing world | 10 |

Source: Authors' contribution.

*4.11. Co-Occurrence of Keywords*

The analysis of the co-occurrence of keywords identifies the presence and frequency of similar keywords used by authors in their research articles. For research in green innovation and financial inclusion, the different keywords used can be seen in Table 8, along with the number of co-occurrences of the particular keywords. It was found that "financial inclusion" is the most used keyword, with the maximum no. of occurrences, i.e., 101, and a total link strength of 91, followed by "fintech" and "sustainable development". Furthermore, "financial inclusion" is used together with "micro-finance", "energy efficiency", and "carbon emissions" (Figure 7). "Green innovation" is used with "digital financial inclusion" and "green finance". The figure shows a research gap between financial inclusion and sustainable finance.

**Table 8.** Occurrences of keywords.

| Keywords | Co-Occurrence |
|---|---|
| Financial inclusion | 101 |
| Fintech | 91 |
| Sustainable development | 85 |
| Sustainability | 72 |
| Digital finance | 66 |
| Sustainable development goals | 54 |
| Financial technology | 52 |
| Economic growth | 46 |
| Digital financial inclusion | 44 |
| Poverty | 39 |

Source: Authors' contribution.

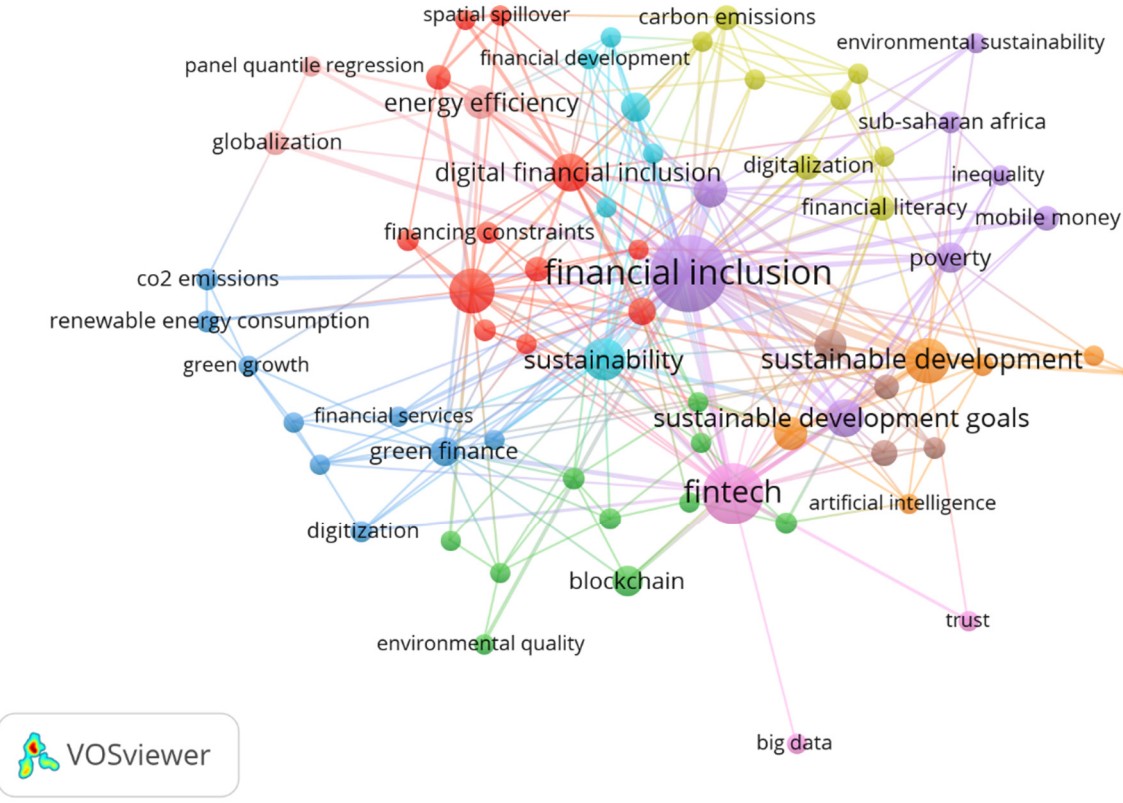

**Figure 7.** A network analysis of keywords. Source: Authors' contribution.

## 5. Limitations

This study also has some limitations that need to be highlighted. As a result of the selection process, many lectures, editorials, book chapters, and notes were omitted, following the adoption of the exclusion criteria, as they did not undergo peer-review processes, which confirm the reliability and good quality of an article. Furthermore, only English-language articles were analyzed during the procedure; again, this sampling bias could have had an influence on the results of our study.

## 6. Conclusions and Policy Implications

Carbon neutrality, as a goal, can be achieved when greener innovations are integrated into the financial inclusion domain. Such innovations help in promoting the use of renewable energy and the upliftment of the underprivileged society for a sustainable future. Therefore, all of the stakeholders, including the government, policymakers, customers, financial institutions, etc., who are involved should prioritize these greener innovation adoptions. Only then do financial inclusion and sustainability go hand in hand.

Climate neutrality is a very ambitious goal to ensure the development of a sustainable future for future generations and all mankind. Careful analysis and screening of the papers revealed the importance of a shared, bottom-up commitment, where society, business, and governments must act in the spirit of collaboration, aimed at promoting an economic and organizational structure that is capable of supporting the spread of new ecological innovations in the energy sector and agricultural–forestry sector.

Based on this literature review, a few future research questions have also been identified, which can help the academic community carry out the work in this research domain. These future research questions are as follows: How can policymakers ensure that green financial inclusion (GFI) initiatives are implemented throughout the countries? What can be the role of microfinance institutions in promoting green financial inclusion? List the pathbreaking technological innovations that can boost the adoption of GFI. List the steps by which countries can ensure that their financial sector is inclusive and sustainable. Establish

a list of enablers and barriers of GFI adoption. How can GFI help in the transmission of economic growth? What strategies can help to promote GFI on a global scale? How are greener innovations in the financial inclusion space shaping a better future through carbon neutrality?

The study examined green innovation practices and innovative financial solutions that are addressing environmental challenges through their promising approaches. The present review article reveals that a lot of potential lies in the advancement of GFI for carbon neutrality by contributing to climate change mitigation, carbon emission reduction, and the achievement of the SDGs. Greener innovations not only drive green entrepreneurship and create green jobs, but they can also stimulate economic growth. Therefore, appropriate policy and regulatory frameworks need to be developed that can incentivize the adoption of GFI practices.

**Author Contributions:** Conceptualization, L.E. and K.S.D.; methodology, L.E., S.A. and K.S.D.; validation, M.B., A.P., A.K.G. and N.T.L.; formal analysis, L.E., K.S.D. and S.A.; writing—original draft preparation, L.E., A.P., K.S.D., S.A. and M.B.; writing-review and editing, M.B., A.P., A.K.G. and N.T.L.; visualization, S.A. and K.S.D.; supervision, M.B., A.P., A.K.G. and N.T.L.; project administration, M.B.; funding, M.B. All authors have read and agreed to the published version of the manuscript.

**Funding:** This research received no external funding. The APC was funded by M.B.

**Institutional Review Board Statement:** Not applicable.

**Informed Consent Statement:** Not applicable.

**Data Availability Statement:** Not applicable.

**Conflicts of Interest:** The authors declare no conflict of interest.

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
