# Peer review of "The Role of Greener Innovations in Promoting Financial Inclusion to Achieve Carbon Neutrality: An Integrative Review"

_economies, doi:10.3390/economies11070194_

Round 1

Reviewer 1 Report

This study investigate the relation between financial technological innovations and environmental projection (climate neutrality). It reviews the recent peer-reviewed aritcle publications and does a literature analysis. I have the following suggestions for the author(s), listing from the most to the least important.

i. The paper pays most attention to the literature structure, i.e., how does the existing literature network with each other. The author(s) did a really great job to clearly present the literature network. However, I find no discussion regarding their results. The author should use some methods (such as meta analysis) to show whether most of the literature finds consistant results or not.

ii. I suggest to drop Section 6. If the future questions are not important, it should be dropped from the content. Oppositely, if the future questions are really important, the author should investigate.

iii. Watermarks in the figures should always be cleared. Instead, in the figure notes the author(s) should mention how was the figure made.

iv. In Fig.2 as the currect barchart is misleading, The author(s) can either drop the bar of 2023, or draw the projected value (i.e, 4* {Value before Mar 31 2023}).

English quality is acceptable, but some changes should be made. Some sentences are not really idiomatically written.

Author Response

Cover Letter for the 1st Reviewer

Title: Role of greener innovations in financial inclusion for the carbon neutrality

Manuscript ID: economies-2400626

Journal: Economies

We are thankful to the editor and anonymous reviewer for their insightful and scholastic comments. The review comments are constructive for improving the article. We have tried our best to address all the review comments in the revised manuscript. All the revision under track changes can been seen in the Red color to mention the additional changes and the revisions in this revised manuscript.

Also, we would like to point out that since the manuscript needed major revision, accordingly we have done changes to enhance its quality. A few comments from reviewers were similar with respect to making the required changes in Introduction. So, we tried to make the changes done in that section with one single color to avoid any confusion.

1st Reviewer:

This study investigate the relation between financial technological innovations and environmental projection (climate neutrality). It reviews the recent peer-reviewed aritcle publications and does a literature analysis. I have the following suggestions for the author(s), listing from the most to the least important.

Comment- The paper pays most attention to the literature structure, i.e., how does the existing literature network with each other. The author(s) did a really great job to clearly present the literature network. However, I find no discussion regarding their results. The author should use some methods (such as meta analysis) to show whether most of the literature finds consistant results or not.

Response- We are thankful to the reviewer for pointing out this. We have tried to cover many aspects of this research theme through three different research questions. We would like to point out that since the methodology of this manuscript is systematic literature review combined with Bibliometric analysis, different relevant sections have tried to answer the consistency of the results. In future, we can think of meta-analysis on this domain as this is picking up the interest.

Comment- I suggest to drop Section 6. If the future questions are not important, it should be dropped from the content. Oppositely, if the future questions are really important, the author should investigate.

Response- We appreciate the wonderful suggestion by the reviewer. We strongly feel that some future scope of work should be mentioned so that the future researchers can make an attempt to explore them. Once the future research questions are known it makes the task easier. Based on the suggestion given, we have tried to drop this section and combine it with the conclusion section.

Comment- Watermarks in the figures should always be cleared. Instead, in the figure notes the author(s) should mention how was the figure made.

Response- We appreciate the wonderful suggestion by the reviewer. As these figures are taken from the analysis of the Bibliometric software (Here R-studio Bibliometrix package), the watermarks are embedded in the images to show the authenticity of the analysis. Absence of such watermarks may also imply that the results have been tampered with or manipulated.

Comment- In Fig.2 as the currect barchart is misleading, The author(s) can either drop the bar of 2023, or draw the projected value (i.e, 4* {Value before Mar 31 2023}).

Response- We are grateful for the minute observations by the reviewer. We have taken into consideration this wonderful suggestion. We have mentioned this in the manuscript Table 1 also. Since the number of articles are quite significant till March 2023 which shows that this domain is picking up interest. Based on many other SLR articles, it has been found that the bar of the recent year (with only few months) cannot be dropped as it highlights the exponential rise of scholarly interest in that area.

Comment- Comments on the Quality of English Language. English quality is acceptable, but some changes should be made. Some sentences are not really idiomatically written.

Response- Thank you. We appreciate the wonderful suggestion by the reviewer. Accordingly, we have made changes.

Reviewer 2 Report

1. What is the main question addressed by the research? The topic of the reviewed article is very current and its implementation contributes to the organization of the current state of knowledge through a carefully prepared literature query.

2. Do you consider the topic original or relevant in the field? Does it address a specific gap in the field? A carefully prepared iterature review provides a compendium of the current state of knowledge, which can be very useful not only for novice researchers, but also for people with more scientific experience.

3. What does it add to the subject area compared with other published material?

As mentioned above, the reviewed material provides primarily a synthetic compendium of the current state of scientific research.

4. What specific improvements should the authors consider regarding the methodology? What further controls should be considered? Analytical methods appropriate for this type of research were used.

5. Are the conclusions consistent with the evidence and arguments presented and do they address the main question posed? Conclusions from the research are formulated in a clear way and are closely related to the obtained results of the analyses. The work is arranged in a logical way, which allows you to easily follow the author's analytical process.

6. Are the references appropriate? The number of the cited bibliographic publications is appropriate and consistent with the current research issues.

7. Please include any assitional comments on the tables and figures. I have no objections to the figures and tables.

Author Response

Cover Letter for the 2nd Reviewer

Title: Role of greener innovations in financial inclusion for the carbon neutrality

Manuscript ID: economies-2400626

Journal: Economies

We are thankful to the editor and anonymous reviewer for their insightful and scholastic comments. The review comments are constructive for improving the article. We have tried our best to address all the review comments in the revised manuscript. All the revision under track changes can been seen in the Red color to mention the additional changes and the revisions in this revised manuscript.

Also, we would like to point out that since the manuscript needed major revision, accordingly we have done changes to enhance its quality. A few comments from reviewers were similar with respect to making the required changes in Introduction. So, we tried to make the changes done in that section with one single color to avoid any confusion.

Comments and Suggestions for Authors

Comment- What is the main question addressed by the research? The topic of the reviewed article is very current and its implementation contributes to the organization of the current state of knowledge through a carefully prepared literature query.

Response- We are grateful for the minute observations about the entire manuscript by the reviewer and are thankful for understanding the importance of this topic.

Comment- Do you consider the topic original or relevant in the field? Does it address a specific gap in the field? A carefully prepared iterature review provides a compendium of the current state of knowledge, which can be very useful not only for novice researchers, but also for people with more scientific experience.

Response- We thank you for your comments.

Comment- What does it add to the subject area compared with other published material? As mentioned above, the reviewed material provides primarily a synthetic compendium of the current state of scientific research.

Response- We appreciate the wonderful suggestion by the reviewer. Accordingly, we have made changes. In the revision stage we have further tried to improve the quality of the manuscript by refining it further.

Comment- What specific improvements should the authors consider regarding the methodology? What further controls should be considered? Analytical methods appropriate for this type of research were used.

Response- We thank you for your comments.

Comment- Are the conclusions consistent with the evidence and arguments presented and do they address the main question posed? Conclusions from the research are formulated in a clear way and are closely related to the obtained results of the analyses. The work is arranged in a logical way, which allows you to easily follow the author's analytical process.

Response- We appreciate your comments. Also, we have tried to improve the manuscript to a great extent in the revision.

Comment- Are the references appropriate? The number of the cited bibliographic publications is appropriate and consistent with the current research issues.

Response-  We thank you for your observations on this aspect.

Comment- Please include any assitional comments on the tables and figures. I have no objections to the figures and tables.

Response- We thank you for your comments for the entire manuscript.

Author Response

Cover Letter for the 3rd Reviewer

Title: Role of greener innovations in financial inclusion for the carbon neutrality

Manuscript ID: economies-2400626

Journal: Economies

We are thankful to the editor and anonymous reviewer for their insightful and scholastic comments. The review comments are constructive for improving the article. We have tried our best to address all the review comments in the revised manuscript. All the revision under track changes can been seen in the Red color to mention the additional changes and the revisions in this revised manuscript.

Also, we would like to point out that since the manuscript needed major revision, accordingly we have done changes to enhance its quality. A few comments from reviewers were similar with respect to making the required changes in Introduction. So, we tried to make the changes done in that section with one single color to avoid any confusion.

Comments and Suggestions for Authors

I want to congratulate the authors for the good work they have done. Although it is a good paper, I still believe that the content could be improved taking into account the following points:

Comment- the contribution of the purposed work is not clearly defined in both the abstract and introduction section, so need to mention it more clear.

Response- We appreciate your wonderful suggestions. Accordingly, we have made changes in the revision stage by streamlining the Introduction. In the revision stage we have further tried to improve the quality of the manuscript by refining it further.

Comment- I verified that there are papers that are not part of the bibliometric analysis and that contain some of the keywords mentioned and were published until March 2023, so they should be included in this analysis.

Response- We are thankful to the reviewer for pointing out this. We have tried to cover papers in the revision stage. However, there are certain articles, which only show up in the search results, but when they are explored properly, they were not found relevant to the topic. Accordingly, we have dropped them in our inclusion criteria. Also, for Literature review articles, it is difficult to cover every article which shows up in the search result. We have tried our best to include the ones which find resemblance to the topic.

Reviewer 4 Report

The paper is interesting, it deals with an important topic of the role of greener innovations in financial inclusion for the carbon neutrality, and it is my pleasure to review it. 

The paper is detailed, well organized, and uses adequate logical tools.

However, I would have some suggestion and recommendation in order to improve the quality of the paper.

The Introduction is long and somewhat confusing, it combines fundamental ideas, establishing the context of the research, with collateral, marginal information, deviating the focus from the main objectives. Also, a presentation of the main chapters of the paper in relation to its objectives could be useful in this context.

How many research questions are issued? On page 5 (Chapter 3.1), the author mentions "A total of four research questions (RQ) were formulated..." but only three RQs are listed below.

Unclear sentences:

-          "Since, Scopus and Web of Science (WoS) data-bases provide the advantage for applying bibliometric methods which prompted the authors (Paul et al., 2021)."

-          "Scopus has a more extensive coverage than the remaining two databases" ...which are the two remaining databases? WoS and ….

Chapter 4.6. Top countries and Affiliation, and the Table 6 are inconsistent and contradictory. If China, the UK and Spain are the countries with the most influential authors (universities?), how does the University of Kinshasa appear (I assume from Democratic Republic of Congo) as the most influential university (affiliation)?

Perhaps, the source of the ambiguity may come from the imprecise and complicated formulation of RQ2: What are the influential articles, authors, journals, countries and affiliations in this area?

In other words, if the question is reasonable for the (most) "influential articles, authors, journals", instead, for "influential... countries and affiliations"... it is confusing .... how is an affiliation "influential"? please explain.

The final Conclusions come somewhat too quickly and consist of general (otherwise correct) statements but superficially related to the results of the present research. What is the clear, synthetic answer to the 3 research questions? How are the results of this research integrated into the international flow of knowledge on these topics?

The statement at the end "The articles confirm that green innovations are becoming a major factor in promoting carbon neutrality and ensuring a transition to a sustainable, low-carbon economy" is somewhat exaggerated, as the most of the analyzed articles focusing on the importance and necessity of this orientation/transition, and even less asserting that "green innovations are becoming a major factor...".

An overall remark: The research and systematization effort are meritorious, but we must not forget that a bibliometric analysis (no matter how sophisticated it may be) is... a bibliometric analysis, with a strong quantitative character... and that's it. Mentioning outstanding works in terms of their bibliometric performance is useful, but cannot replace reading them, and especially understanding them in context. Otherwise, we move into an obsession with rankings and scores, which is not useful for the scientific endeavour.

Returning to the title of the article - from this perspective, it does not seem to fully suggest the content of the article, being more of a bibliometric, thematic and citation analysis of the role of greener innovations in financial inclusion for the carbon neutrality... A revision of the title in this sense is recommended.

Formal :  Abbreviations are explained when they first appear in the text. Then they can be used as such, see ESG, SDG, GFI.

Thank you for the opportunity to review this article and good luck!

Author Response

Cover Letter for the 4th Reviewer

Title: Role of greener innovations in financial inclusion for the carbon neutrality

Manuscript ID: economies-2400626

Journal: Economies

We are thankful to the editor and anonymous reviewer for their insightful and scholastic comments. The review comments are constructive for improving the article. We have tried our best to address all the review comments in the revised manuscript. All the revision under track changes can been seen in the Red color to mention the additional changes and the revisions in this revised manuscript.

Also, we would like to point out that since the manuscript needed major revision, accordingly we have done changes to enhance its quality. A few comments from reviewers were similar with respect to making the required changes in Introduction. So, we tried to make the changes done in that section with one single color to avoid any confusion.

Comments and Suggestions for Authors

The paper is interesting, it deals with an important topic of the role of greener innovations in financial inclusion for the carbon neutrality, and it is my pleasure to review it. The paper is detailed, well organized, and uses adequate logical tools. However, I would have some suggestion and recommendation in order to improve the quality of the paper.

Comment: The Introduction is long and somewhat confusing, it combines fundamental ideas, establishing the context of the research, with collateral, marginal information, deviating the focus from the main objectives. Also, a presentation of the main chapters of the paper in relation to its objectives could be useful in this context.

Response: We thank the reviewer for the critical observation. We have improved our Introduction section to a great extent based on the suggestions.

Comment: How many research questions are issued? On page 5 (Chapter 3.1), the author mentions "A total of four research questions (RQ) were formulated..." but only three RQs are listed below.

Response : Thank you for pointing this out. The corrections have been made in the manuscrit and highlighted.

Comment: Unclear sentences- "Since, Scopus and Web of Science (WoS) data-bases provide the advantage for applying bibliometric methods which prompted the authors (Paul et al., 2021)."

Response : The corrections have been made in the manuscrit and highlighted.

Comment: "Scopus has a more extensive coverage than the remaining two databases" ...which are the two remaining databases? WoS and ….

Response : The corrections have been made in the manuscrit and highlighted.

Comment: Chapter 4.6. Top countries and Affiliation, and the Table 6 are inconsistent and contradictory. If China, the UK and Spain are the countries with the most influential authors (universities?), how does the University of Kinshasa appear (I assume from Democratic Republic of Congo) as the most influential university (affiliation)?

Perhaps, the source of the ambiguity may come from the imprecise and complicated formulation of RQ2: What are the influential articles, authors, journals, countries and affiliations in this area?

In other words, if the question is reasonable for the (most) "influential articles, authors, journals", instead, for "influential... countries and affiliations"... it is confusing .... how is an affiliation "influential"? please explain.

Response: The result of top ten countries, authors, journals and affliations have been extracted from the Scopus database and the information was analysed by the Scopus database. In this case, it can be viewed that the countries and affliations can differ according to the multiple authors in the documents which creates the difference among countries and their affilations. Also, we have corrected the sentence. Now instead of inflential journal, countries and affliation it is now the most productive journal and most contributing journals and affliations that are publishing the articles in this domain. Hence, the data is not ambigious, it contains the correct information.

Comment: The final Conclusions come somewhat too quickly and consist of general (otherwise correct) statements but superficially related to the results of the present research. What is the clear, synthetic answer to the 3 research questions? How are the results of this research integrated into the international flow of knowledge on these topics?

Response: We are grateful to you for pointing out this. We have made changes in the conclusion.

Comment: The statement at the end "The articles confirm that green innovations are becoming a major factor in promoting carbon neutrality and ensuring a transition to a sustainable, low-carbon economy" is somewhat exaggerated, as the most of the analyzed articles focusing on the importance and necessity of this orientation/transition, and even less asserting that "green innovations are becoming a major factor...".

Response: Thank you for the suggestion. We have made relevant changes to the conclusion section.

Comment: An overall remark: The research and systematization effort are meritorious, but we must not forget that a bibliometric analysis (no matter how sophisticated it may be) is... a bibliometric analysis, with a strong quantitative character... and that's it. Mentioning outstanding works in terms of their bibliometric performance is useful, but cannot replace reading them, and especially understanding them in context. Otherwise, we move into an obsession with rankings and scores, which is not useful for the scientific endeavour.

Response: Thank you for pointing out this, we have made relevant changes across the entire manuscript to highlight the importance of this article and its scientific contribution to the academic research.

Comment: Returning to the title of the article - from this perspective, it does not seem to fully suggest the content of the article, being more of a bibliometric, thematic and citation analysis of the role of greener innovations in financial inclusion for the carbon neutrality... A revision of the title in this sense is recommended.

Response: Thank you for pointing out to bring the change in the title. We have modified it. – ‘The role of greener innovations for promoting the financial inclusion to achieve carbon neutrality: An integrative review’

Comment: Formal :  Abbreviations are explained when they first appear in the text. Then they can be used as such, see ESG, SDG, GFI. Thank you for the opportunity to review this article and good luck!

Response: The corrections have been made in the manuscrit and highlighted.

Round 2

Reviewer 1 Report

The authors have made corresponding changes and I have no more questions.

Reading flows are fine to me.

Reviewer 3 Report

The changes in my opinion are sufficient for the paper to be published. I am in favor of its publication.

No English proofreading is required.

Reviewer 4 Report

In this current version of the manuscript, the authors addressed all our suggestions and recommendations, significantly improving the calrity and the value of the study. Thus, the paper is more suitable for publication.